# Holographic Acoustic Tweezers for 5-DoF Manipulation of Nanocarrier Clusters toward Targeted Drug Delivery

**DOI:** 10.3390/pharmaceutics14071490

**Published:** 2022-07-18

**Authors:** Hiep Xuan Cao, Daewon Jung, Han-Sol Lee, Van Du Nguyen, Eunpyo Choi, Byungjeon Kang, Jong-Oh Park, Chang-Sei Kim

**Affiliations:** 1School of Mechanical Engineering, Chonnam National University, Gwangju 61186, Korea; hiep.caoxuan@kimiro.re.kr (H.X.C.); hansol3607@gmail.com (H.-S.L.); nvdu81@gmail.com (V.D.N.); eunpyochoi@jnu.ac.kr (E.C.); 2Korea Institute of Medical Microrobotics, Gwangju 61011, Korea; jungdaewon@kimiro.re.kr (D.J.); bjkang8204@jnu.ac.kr (B.K.); jop@kimiro.re.kr (J.-O.P.); 3College of AI Convergence, Chonnam National University, Gwangju 61186, Korea

**Keywords:** acoustic manipulation, acoustic tweezers, nanocarrier clusters

## Abstract

Acoustic tweezers provide unique capabilities in medical applications, such as contactless manipulation of small objects (e.g., cells, compounds or living things), from nanometer-sized extracellular vesicles to centimeter-scale structures. Additionally, they are capable of being transmitted through the skin to trap and manipulate drug carriers in various media. However, these capabilities are hindered by the limitation of controllable degrees of freedom (DoFs) or are limited maneuverability. In this study, we explore the potential application of acoustical tweezers by presenting a five-DoF contactless manipulation acoustic system (AcoMan). The system has 30 ultrasound transducers (UTs) with single-side arrangement that generates active traveling waves to control the position and orientation of a fully untethered nanocarrier clusters (NCs) in a spherical workspace in water capable of three DoFs translation and two DoFs rotation. In this method, we use a phase modulation algorithm to independently control the phase signal for 30 UTs and manipulate the NCs’ positions. Phase modulation and switching power supply for each UT are employed to rotate the NCs in the horizontal plane and control the amplitude of power supply to each UT to rotate the NCs in the vertical plane. The feasibility of the method is demonstrated by in vitro and ex vivo experiments using porcine ribs. A significant portion of this study could advance the therapeutic application such a system as targeted drug delivery.

## 1. Introduction

Acoustic tweezers have demonstrated potential for use in medical applications, particularly for the targeted delivery of cells, compounds, or living things, from nanometer-sized extracellular vesicles to centimeter-scale particles. They are capable of being transmitted through the skin to trap and manipulate drug carriers in various media.

Rayleigh et al. [1,2] first proposed the idea of manipulating objects using acoustic radiation and provided a theoretical background for the same in 1903. In 1923, Boyle et al. [3], pioneers in the development of modern ultrasound, experimentally demonstrated for the first time that bubbles of various sizes could be produced and trapped in volatile organic liquids (e.g., benzol) using acoustic energy. In 2015, Marzo et al. [4] demonstrated how polystyrene particles ranging from 0.6 to 3.1 mm in diameter levitating above single-sided arrays can be manipulated in three dimensions and rotated along the vertical axis in air using acoustic tweezers at 40 kHz. Acoustic tweezer uses the acoustic radiation force exerted by the interaction of sound waves with solids, liquids, and gases to trap particles with sizes in the micrometer-sized to centimeter-sized without contact. They have become a versatile tool in disease diagnosis [5], cell manipulation [6,7], and living body applications, such as the manipulation of objects in the bladder [8]. In recent studies [9,10,11,12,13,14,15], acoustic tweezers were grouped in three categories according to their working principles: (i) standing-wave, (ii) traveling-wave, and (iii) acoustic streaming tweezers. The former two classes of tweezers are direct methods that apply external acoustic radiation forces to manipulate objects, whereas acoustic streaming tweezers employ medium flow (e.g., fluid, air) to control object orientation [16,17]. Acoustic streaming tweezers combine acoustics and microfluidics to indirectly manipulate particles via acoustically induced fluid flows and require microfluidic channels to activate the stream of fluid flow [18,19]. Acoustic tweezers based on standing wave utilizes interdigitated transducers (IDTs) [20] and generally used for separating and patterning of particles and cells inside the channels. On the other hand, traveling-wave tweezers can create arbitrary pressure nodes in three-dimension space using a single element or multi-elements in real-time. Recently, traveling-wave tweezers have become attractive methods to trap and manipulate an object for biomedical applications [4,8,21,22].

To become a versatile and efficient platform for the scientific and medical communities, suitable structural configurations for medical applications and the number of controllable degrees-of-freedom (DoFs) of acoustic tweezers must be improved. In this study, we explored the capabilities of holographic acoustic tweezers to perform a five-DoF contactless manipulation (three DoFs for position and two DoFs for pointing orientation) of nanocarrier clusters (NCs) in water with UT array with single-side arrangement. NCs can be manipulated in five DoFs along a large workspace, which has not been achieved in previous studies. Table 1 summarizes the differences between the proposed system and recently developed systems. The proposed system is the first system that can manipulate objects in five DoFs in water. With a single-side arrangement, the system is suitable for an in vivo environment and can target different parts of the human body.

AcoMan is an acoustic tweezer platform designed for this study. The single-side arrangement of AcoMan can maximize acoustic pressure. Thus, the intensity of acoustic radiation force is maximum at the focal point compared with other arrangements.

By only controlling the phase offset to each transducer, NCs can move through a 6 mm × 6 mm × 5 mm working space at any point, rotate completely unrestrained in the O-XY plane with a 45° step angle resolution, and rotate in the O-XZ plane with a maximum angle of −30.31° in the counterclockwise direction. Increasing the number of transducers in array reduces the step angle resolution for rotating in the O-XY plane. By increasing the difference in amplitude of power for each side of AcoMan, the maximum angle of NCs plane with respect to the O-XZ plane can be increased. AcoMan was designed for underwater conditions, which have acoustic impedance close to the skin and tissue. In the future, AcoMan will be developed for an advanced targeted drug delivery technique in the living body. Figure 1 shows the conceptual image of the proposed system for controlling micro/nano carriers for targeted drug delivery in medical applications.

In targeted drug delivery, a carrier must deliver a drug payload through blood vessels with a diameter from micrometers to millimeters to a specific region. Hence, the size of the carrier and amount of payload carried by the carrier are limited [28]. To address this limitation, the carrier is aggregated into chains or clusters that can move in a group to reach the target with a large amount of payload. In addition to clustering, this structure can change the angle of the entire structure. Thus, the contact area between the cluster and target can be decided by controlling the angle of the cluster once it reaches the target. This approach allows the drug to reach the target more efficiently.

The study is organized into the following four sections. Section 2 details the materials, methods, model of acoustic radiation force, design and fabrication of AcoMan, and method for controlling nano/micro carriers in five DoFs. Section 3 provides the results of an in vitro experiment based on the proposed method. Finally, Section 4 discusses the research presented and concludes the study.

## 2. Materials and Method

### 2.1. Model of Acoustic Radiation Force

To calculate the acoustic field, a single frequency far-field piston using acceleration control was employed for each transducer. The complex acoustic pressure P generated by the array at a point is the vector sum of 30 complex acoustic pressures *P_j_* due to a piston source emitting at a single frequency, which is expressed as Equation (1) [29]:(1)$$\overset{30}{\underset{j=1}{\sum }}P_{j}(r)=P_{0}A\frac{D_{f}\left(\theta \right)}{d}e^{i(\varphi +kd)},$$
where $P_{0}$ is a constant that defines the transducer output efficiency, A is the peak-to-peak amplitude of the excitation signal, and $D_{f}\left(\theta \right)$ is a far-field directivity function that depends on the angle *θ* between the perpendicular line to the transducer surface and the point r, the term 1/d  corrects for divergence, where d is the propagation distance in free space, k=2π/λ is the wavenumber, and *φ* is the phase of the emitting piston source. For a single transducer acting as a circular piston source, the ratio of the sound wavelength λ to the size *a* decides its directivity. Hence, the directivity function [30,31] can be written as,
(2)$$D_{f}=\frac{2J_{1}\left(kasin\theta \right)}{ka\sin\theta },$$
where $J_{1}$ is a first-order Bessel function of the first kind, a is the radius of the emitting source. Additionally, the phase spreading of the piston source *j* determines the manipulation strategy method applied to generate a trap in the acoustic fields. The twin and vortex traps are the most widely used phase modulation methods of acoustic tweezers. In the vortex trap, the target agent spins around its axis without orientation control. On this aspect, we choose the phase modulation strategy to create a twin trap, which uses a twin signal distribution with a phase of *φ* = 0° or 180° that can control the orientation.

When the acoustic pressure field is formed in the acoustic region, the most dominant force used to manipulate the particles in the region of interest is the acoustic radiation force (ARF). The ARF exerted on a small spherical particle (${\vec{F}}^{rad}$) can be calculated based on the gradient of the Gor’kov potential field [32,33], U.
(3)$${\vec{F}}^{rad}=-\nabla U,$$
(4)$$\nabla^{2}U=\frac{\partial^{2}U}{\partial^{2}x}+\frac{\partial^{2}U}{\partial^{2}y}+\frac{\partial^{2}U}{\partial^{2}z},$$
(5)$$U=2K_{1}\left({\left|p\right|}^{2}\right)-2K_{2}\left({\left|p_{x}\right|}^{2}+{\left|p_{y}\right|}^{2}+{\left|p_{z}\right|}^{2}\right),$$
(6)$$K_{1}=\frac{1}{4}V\left(\frac{1}{c_{0}^{2}\rho_{0}}-\frac{1}{c_{1}^{2}\rho_{1}}\right),$$
(7)$$K_{2}=\frac{3}{4}V\left(\frac{\rho_{0}-\rho_{1}}{\omega^{2}\rho_{0}\left(\rho_{0}+2\rho_{1}\right)}\right),$$
where V is the volume of the spherical particle, ω is the angular frequency of the emitting source acoustic wave, ρ is the density, and c is the speed of sound (with subscripts 0 and 1 referring to the host medium and the particle material, respectively). Finally, p is the complex pressure and its derivative over the X-, Y-, and Z-directions in the Cartesian axes are $p_{x}$, $p_{y}$, and $p_{z}$, respectively. The density and speed of sound of nano particle material are 1032 kg/m^3^ and 1489–1499 m/s, respectively [34,35]. The density and speed of sound of water are 997 kg/m^3^ and 1045 m/s.

### 2.2. Design and Fabrication of AcoMan

We focused on the requirements of a system suitable for human applications. Ultrasound waves propagate through tissue as traveling waves. High-frequency ultrasound waves with shorter wavelengths can manipulate smaller particles but have limited penetration depth. Conversely, low-frequency ultrasound can penetrate more deeply but with larger particle manipulability. The system was immersed in water because the acoustic impedance of water (1480 × 10^3^ kg/s·m^2^) is relatively similar to that of human tissue (1530 × 10^3^ kg/s·m^2^) [36]. The array was designed to generate maximum acoustic pressure at the focal points with 30 transducers. As illustrated in Figure 2a–c, the UTs array was designed as a single-side shape with 30 UTs in three circular layers. The design specifications of AcoMan are summarized in Table 2.

We designed and fabricated AcoMan, which comprises three main subsystems: (1) the UTs array, (2) the signal amplifier, and (3) the software control interface. The array comprised 30 immersible ultrasonic transducers (10 mm diameter, NI101, Japan Probe, Yokohama, Japan). The UTs were arranged in a single-side shape and positioned in three circular layers, as shown in Figure 2a, to generate maximum acoustic pressure at the focal point with the number of transducers. Thus, the strongest acoustic radiation force was applied to the object at the focal point [33]. To drive the UTs at 1 MHz, we used a customized amplifier that we reported in our previous work [22,37] with 16 additional channels and an independent power control module for each group of four transducers. The square wave signals generated by the NI PCIe-7852R with LabVIEW FPGA 2017 module at 3.3 Vpp contained the phase and amplitude of power information. The signals were amplified to 200 Vpp while keeping the phase information to drive the transducer. In this study, we amplified the signal in the 10–100 Vpp range and operated the transducer using bipolar voltage.

AcoMan required three power sources with different amplitudes: DC 5 V and DC 3.3 V. The power was supplied by a K6333A-regulated DC power source (EXSO, Korea) with a maximum current of 3 A and accuracy of ± 0.1 A. The bipolar high voltage side from ±5 V to ±50 V was supplied in series by an LPS-503TP-regulated DC power source (DNDE, Korea) with a maximum current of 2 A and accuracy of ±0.1 V. Next, a software control interface that allows users to control the position and orientation of micro/nan motor with real-time image tracking was developed based on the LabVIEW 2017 platform. The software applied close-loop control with position information feedback from two charge-coupled device (CCD) cameras (2.3 MP color blackfly PoE GigE C-mount) mounted at the top and on the side of the setup. Each camera was mounted with a Kowa CCTV lens LMZ45T3 2/3”, F2.5 with a working distance of 18–108 mm. The frame size was set as 680 × 480 pixels for a fast-processing time in the LabVIEW Vision module. The entire system was controlled by LabVIEW 2017 by an Intel core i5 3.2 GHz processor running on Windows 10.

### 2.3. 3D Manipulation of the Nanocarrier Clusters

To allow 3D manipulation of the NCs, tweezer-like twin traps were generated by adjusting phase delays of the driving signal to each element in the array. The phase delays of the twin trap include both the sum of focus phase information and the twin traps phase information, which was calculated based on the definition of two sides with a π-phase difference between them in array geometry. The focus phase information was calculated by finding the position of each transducer in the array relative to the position of the desired focal point, which creates a single beam. To do that, the number of acoustic wave cycles (N) was calculated as follows:(8)$$Number\;of\;cycles\;\left(N\right)=\frac{\sqrt{{\left(P_{X}-T_{Xi}\right)}^{2}+{\left(P_{Y}-T_{Yi}\right)}^{2}+{\left(P_{Z}-T_{Zi}\right)}^{2}}}{\lambda },$$
where the desired focal position is at *P*$\left(P_{X},\;P_{Y},\;P_{Z}\right)$, *T(*$T_{Xi},\;T_{Yi},\;T_{Zi}$) is the individual element position in UT, and λ is the wavelength of acoustic wave.

Subsequently, the phased delay value was calculated for each transducer element:(9)Phase delay (deg)=360·(1−T*(N)),
where the function of *T**(*N*) defines the decimal part of the number of acoustic wave cycles (N) in Equation (7) and (1−T*(N)) is the margin of a last cycle at the derided focal point.

The simulation for the proposed UTs array model and 3D manipulation control method was performed in COMSOL Multiphysics with a resonance frequency at 1 MHz for the 30 transducers. The result of shifting the twin trap point is shown in Figure 3. We electrically generated the twin trap point at (0, 0, 0) and subsequently regenerated it at (1, 1, 1). The acoustic radiation force at twin trap points shown in Figure 3a for (0, 0, 0) and Figure 3b for (1, 1, 1) in -*X*, -*Y*, -*Z* axes were normalized by the maximum pressure along with *X*-axis.

In addition, we implemented the closed-loop control method to improve the 3D manipulation performance. Here, we used a proportional control (P-control). For the position feedback, a CCD camera was mounted on the top captures the position of NCs in the O-XY plane, and another CCD camera was mounted on the side captures the position of NCs in the O-XZ plane. The 3D current position of NCs was obtained by image processing using the LabVIEW Vision module. The position difference between the current and the desired position is compared to calculate the position error in each direction. Error X, Error Y, and Error Z represent the position errors along each axis between the current position and the desired position of the NCs. The threshold of error in each axis is set at 300 μm. After processing each loop of the program with 40 ms sampling rates, the phase value of each transducer was recalculated. Furthermore, the twin trap was reformed to minimize the position error between the desired path and the tracked position of the NCs.

### 2.4. 2D Rotation of the Nanocarrier Clusters

Figure 2d,e show the characteristics of the twin- trap comprising a tweezer with two cylindrical beams. The initial cylindrical beams can be formed in the O-XZ plane or O-YZ plane. In this study, the initial cylindrical beam was formed in the O-XZ plane. To control the rotation in the O-YZ plane, the initial cylindrical beams should be generated in the O-YZ plane. The system can only rotate the NCs in the O-XY and O-XZ or O-XY and O-YZ planes at the same time. Thus, the system can only achieve 2D rotation control.

First, the rotation of NCs was demonstrated in the O-XY plane (Yaw motion) by employing the phase delay modulation method. Rotating the twin trap signal between the two halves of the array allowed the tweezer structure to be rotated in the same direction. The proposed array with 30 transducers could be divided into eight twin trap signals references, as shown in Figure 4a–h. Thus, the rotation for each step was 360° ÷ 8 = 45°. The rotation step decreased as the number of transducers increased.

Second, the rotation was demonstrated on the O-XZ plane or O-YZ plane (roll motion) with a reference coordinate of array. Each side of tweezer was excited with a different power supply. The custom switching power circuit was made to switch power to each side of the array. This method allowed us to control the excitation voltage reaching each side of the tweezer to within the 10–60 Vpp range independently. We found that the cylinder in the twin trap could go up or down following the excitation voltage. This phenomenon is ideal for reconfiguring the two cylinders in the tweezer to rotate trapping objects along the *X*-axis. Based on the proposed method, we ran the simulation in COMSOL Multiphysics with the excitation voltage different between two side of tweezer from 10 V to 50 V. Figure 5a,b show the normalized acoustic field and radiation force value at the physical focal point of the proposed array (0, 0, 0) with reference to the case excitation voltages for both sides of the system are 60 Vpp. Till date, this method has not been reported, theoretically or experimentally, in both acoustics and optics.

### 2.5. Calibration

A calibration procedure is necessary to ensure that the generated acoustic field agrees with the simulation result. To calibrate AcoMan, we used a 1 mm needle hydrophone (Precision Acoustics, Dorchester, UK) to measure the waveform, and then, used Fourier transform to obtain the phase of each transducer with the reference phase at 0° at the focal point. The phases of the 30 transducers were generated one by one at the same phase value. The phase calibration for each transducer was calculated as follows:(10)$$\varphi^{i}{}_{cal}=\varphi^{i}{}_{mea}-\varphi^{i}{}_{ref}$$
where *φ^i^_ref_* is the reference phase at 0°, *φ^i^_mea_* is the measured phase value in degrees at the physical focal point, and *φ^i^_cal_* is the calibration phase value for transducer *i*, where *i* varies from 1 to 30. Ideally, the phases of all transducers at the focal point should be equal. Herein, the phase difference remained unchanged upon applying a different voltage. The transducer fabrication process and misalignment position into the array during the system assembly were the primary sources of error. Using the measured calibration values for the compensation of phase for the preset value on the LabVIEW FPGA program, the phases of all the transducers signal were the same at the focal point after calibration. To validate that calibration method, we scanned acoustic pressure in twin trap control before and after calibration in the same operating condition. Figure 6a shows the scanned acoustic pressure value at 20 Vpp before calibration with maximum value at 301 kPa and twin trap point at (0.2, −0.4). Figure 6b shows the scanned acoustic pressure value at 20 Vpp after calibration with maximum value at 403 kPa and twin trap point at (0, 0). Figure 6c shows the normalized acoustic pressure field in the simulation with respect to maximum pressure. We can compare the maximum value in Figure 6a,b from the scanned results. It shows that the maximum acoustic pressure after calibration was 25% higher than that before calibration. The acoustic pressure field map is agreed with acoustic pressure field map in the simulation.

### 2.6. Biocompatibility of the Nanocarier Cluster

NCs with a diameter of 100 nm designed for a drug delivery system were used for the experimental demonstration. The adopted fabrication process was based on the solvothermal method reported in [38]. Two kinds of mice cells, 4T1 (mice breast cancer cell line) and NIH3T3 (mice fibrous cell line) cells, were used to investigate the biocompatibility of the NCs. Figure 7a shows the transmission electron microscopy (TEM) images of the nanoclusters used with an average diameter of 98 nm. The results revealed that more than 90% of the cells in all cases were viable; an increase in the concentration of NCs (0–200 μg/mL) decreased the cell viability. Figure 7b show the NCs plane definition in the {O} coordinate system.

## 3. Experimental Results

### 3.1. Experimental Setup

An experiment to investigate control of NCs in five DoFs for in vitro and ex vivo trials was conducted. For the in vitro experiment, we demonstrated 3D position control and 2D rotation control. First, we demonstrated the trapping and manipulation of NCs through the phase modulation method. Next, we performed rotation control in the O-XY and O-XZ planes. Finally, we demonstrated full automated control of NCs in five DoFs following two pre-programmed paths. For the ex-vivo experiment, we performed 3D position control and head-and-go motion in the O-XY plane above porcine ribs. The UTs array was designed using 30 ultrasonic transducers immersed in an acrylic box (300 mm × 300 mm × 300 mm) filled with deionized water. The dimensions for the acrylic box were chosen to approximate the size of the abdominal or thoracic cavity.

To further investigate the performance of the system under open-loop and close-loop control, we used two pre-programmed paths of the device; herein, automated control refers to close-loop control for position and open-loop control for orientation. The experimental setup is shown in Figure 8.

### 3.2. 3D Manipulation of the Nanocarrier Clusters

Three-dimensional manipulation of NCs was accomplished by applying the phase modulation method. First, the NCs is trapped at the initial twin trap point. In addition, the AcoMan controller sends the phase value that can generate the twin trap point at the desired position. After that, NCs moves to the new twin trap point. Where the distance between two twin trap points can be defined as a single-step movement. The distance between two cylindrical beams in the tweezer is 1.5 mm as shown in Figure 3, which is the maximum displacement for each step movement. The smallest displacement is 37.5 μm by considering controller specification. For the specified sampling time, the longer single-step movement will result in higher speed but make a bigger position error and a shorter single-step movement will result in lower speed but can achieve more accurate movements. We searched the optimal single-step distance by trial and error in the experiment, and then, each step movement was determined to be 0.2 mm where the manipulation speed in each single-step movement was 1.89 mm/s.

Two experiments were conducted to demonstrate the performance of 3D manipulation. The first experiment was conducted to control the NCs and find a controllable working space with a position error of less than 300 μm. The NCs moved from the physical focal point (0, 0, 0) to the edge of the working space at (2.5, 2.5, 0) along two elliptical trajectories, generating a globular shape on the working space. The second experiment was conducted to create a trajectory with a pyramid shape. The NCs can reach any point from the physical focal point to the edge of the working space. Based on both the simulations and experimental demonstration as shown in Figure 9, the NCs can be manipulated in 3D to any point in the working space (5 × 5 × 4 mm^3^) with 200 μm displacement steps. The maximum velocity of the NCs during the experiment was approximately 5 mm/s, whereas the maximum position errors for the two experiments were limited under 200 μm by applying a simple proportional-derivative controller with position feedback from two cameras.

### 3.3. 2D Rotation of the Nanocarrier Clusters

The rotation of NCs in the O-XY plane was observed from above the array. As shown in Figure 10a,b, the NCs were rotated from 0° to 360° with 45°-angle steps. The results revealed that the entire cluster was rotated following the input angle control both in the counterclockwise and clockwise directions (the rotation frequency was approximately 3.2 rps). The rotation angle errors were less than 20° at each step and were reset after each step. Thus, the rotation error remained less than 20° independent of the total rotation angle.

Second, the rotation of the NCs along the *X*-axis was demonstrated in the XOZ plane and viewed from the front of the array. The result reveals that the maximum angle was achieved at −30.31° and 29.93° in the counterclockwise and clockwise directions, respectively, with the same voltage difference at 50 Vpp between the two sides of the twin trap. To further demonstrate the performance of the rotation control, we rotated the NCs to reach the maximum angle at each direction and performed the head-and-go motion while keeping the rotation angle constant. The time-lapse image of motion is shown in Figure 11a. The measured angle result shown in Figure 11b. Each demonstration is conducted 3 times.

### 3.4. Ex Vivo Validation

One of the targeted real-life medical applications for AcoMan is drug delivery inside organs under the ribs. The real-life organ environment is complex, it consists of heterogeneous, non-rigid structures with different acoustic impedance properties [39,40]. A challenge encountered in acoustic manipulation in an ex vivo environment is the difference in permeability of the acoustic wave, which decreases as it moves through porcine fat, muscle tissue, and bone. To check the feasibility of this targeted application, an ex vivo experiment was conducted using a 30 mm thick porcine rib with an area of 120 mm × 100 mm, which was sufficient to cover the surface of the UT. The temperature at the focal point was measured by a thermometer (MTM-380SD, Lutron Instruments) during the following operating period at 5 min, 10 min, and 15 min to check the heating effect. The temperature rose from 24.4° to 27.6°. The temperature at the focal point was not over 27.6° during the operation period. First, the cluster was manipulated in the XOZ plane following the preprogrammed square path O→A→B→C→D→O, as shown in Figure 12d. Next, the rotation of the NCs was performed and the manipulation in the O-XY plane followed a triangular path. At the focal point at O’, the cluster rotated 90° counterclockwise and then moved toward A’. At A’, the cluster rotated 135° clockwise and then moved to B’. At B’, the cluster rotated 90° counterclockwise and then moved to C’. At C’, the cluster rotated 45° then moving with tail head to O’. The ex vivo demonstration revealed that NCs as drug carriers were steadily trapped at the trapping point, and the cluster was manipulated in the 4 mm × 4 mm working space in the XOZ plane. The NCs performed a head-and-go motion while rotating and moving in a triangular path on the XOY plane.

## 4. Discussion and Conclusions

In this study, we developed a system that performed a 5-DoF manipulation of NCs using holographic acoustic tweezers called AcoMan for the first time. AcoMan consists of (1) a UTs array with a single-side design, which shows potential for use in in vivo experiments; (2) a signal amplifier, which can amplify the ultrasound signal in the wide range of frequency from 1 Hz to 40 MHz and high-voltage range from 10 Vpp to 220 Vpp; and (3) a user interface on the controller computer that was developed with a modular design that is easy for clinicians to operate.

The system generated stable 3D-holographic tweezers that could capture and manipulate 3D precise positioning under closed-loop control and 2D rotation control under open-loop control. The 3D position control under closed-loop control could manipulate the NCs along the predefined trajectory inside the water with a position error of less than 200 μm. The 2D rotation control, which has a rotation step of 45°, had a rotation angle with no limits along the *Z*-axis and rotation angle ranging from –30.31° to + 29.93° along the *X*-axis. As a proof of concept, AcoMan was used to perform an ex vivo 3D manipulation of NCs as the main drug carrier in porcine ribs with a working space of 5 mm × 5 mm ×4 mm. This study will enable and guide further studies on a new target drug delivery system using acoustic actuators.

Although the currently proposed mechanism can perform 5-DoFs target localization, it has some limitations for application at clinical sites. The experiments were conducted without considering the flow conditions in real organs. The distance from the surface of the array to the controlled target was 75 ± 4 mm, which is too small for application of the system in human bodies. This limitation can be solved by increasing the number of the transducer in the array [9]. In this study, we are only the use the 30 transducers for the organism target, which is close to the array surface, as a proof of concept. The second limitation of AcoMan is the small working space of 5 mm × 5 mm × 4 mm. Further additional mechanical device may resolve this issue. In addition, the rotational angle resolution is a limitation of the proposed system. This needs to be conquered by adding more transducers or additional mechanism in future study. Finally, for the application of the proposed closed-loop control scheme, a camera feedback should be changed to an in vivo compatible method. Based on our recent work that the commercial ultrasound probe (L12) and imaging system (SIEMENS Acuson Sequoia) with 6.5 MHz~12.3 MHz frequency could detect our NCs, an ultrasound imaging system will be further studied for the closed loop control application in in vivo. In addition, animal study as well as aggregation and disaggregation of the nanoparticles by using the proposed system will also be studied for the performance validation toward the targeted drug delivery.

## Figures and Tables

**Figure 1 pharmaceutics-14-01490-f001:**
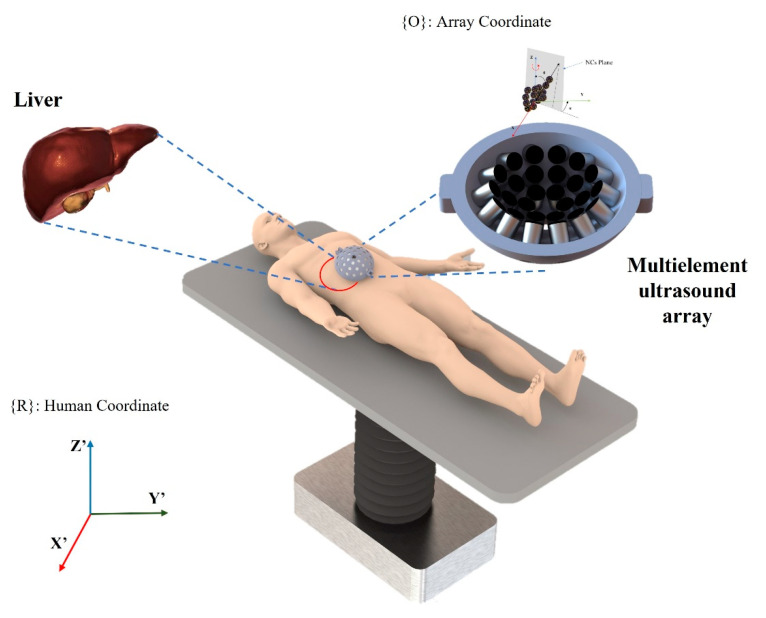
Concept image of AcoMan for controlling micro/nano carriers for targeted drug delivery in the liver. Nanoparticles are aggregated into clusters at trapping position so that they can move in unison; subsequently, they are driven in five DoFs by UTs array. Drugs are released by an external stimulus after the NCs reach the target area.

**Figure 2 pharmaceutics-14-01490-f002:**
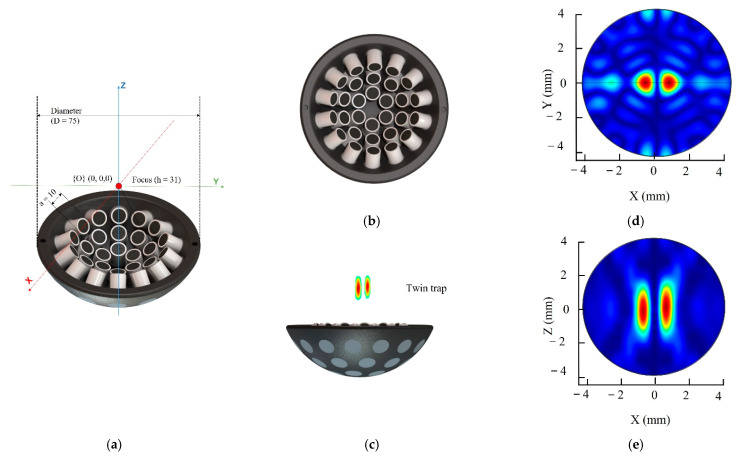
Single-side UTs array geometry design: (**a**) isometric view; (**b**) lateral views; and (**c**) top view; (**d**) Absolute acoustic radiation pressure field in the O-XY plane; (**e**) Absolute acoustic radiation pressure field in the O-XZ plane.

**Figure 3 pharmaceutics-14-01490-f003:**
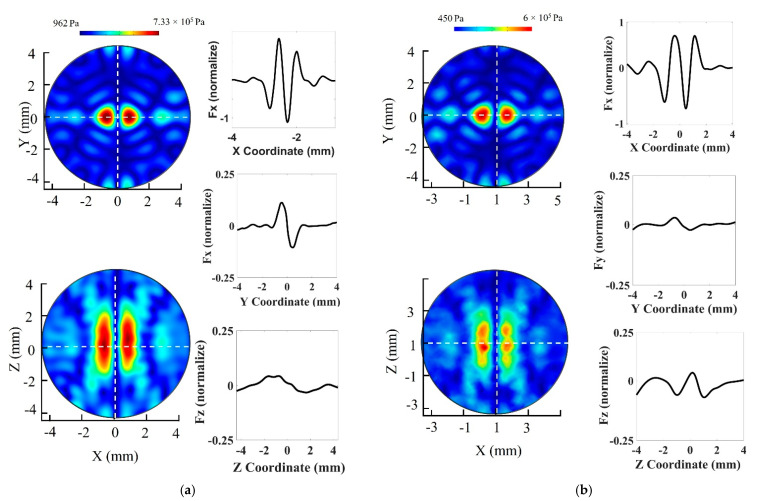
3D manipulation in COMSOL Multiphysics: acoustic pressure field map and normalized acoustics radiation force at twin trap point: (**a**) (0, 0, 0); (**b**) (1, 1, 1).

**Figure 4 pharmaceutics-14-01490-f004:**
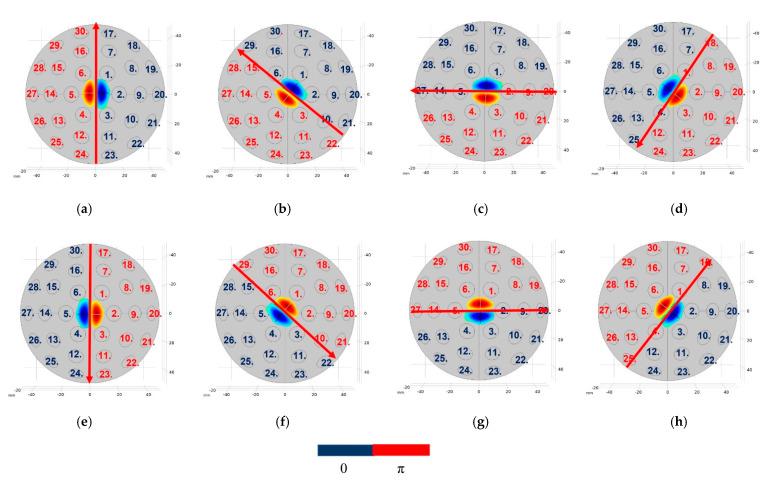
Twin phase information for each transducer in the array rotated along the *Z*-axis: (**a**) 0°; (**b**) 45°; (**c**) 90°; (**d**) 135°; (**e**) 180°; (**f**) 225°; (**g**) 270°; and (**h**) 315°.

**Figure 5 pharmaceutics-14-01490-f005:**
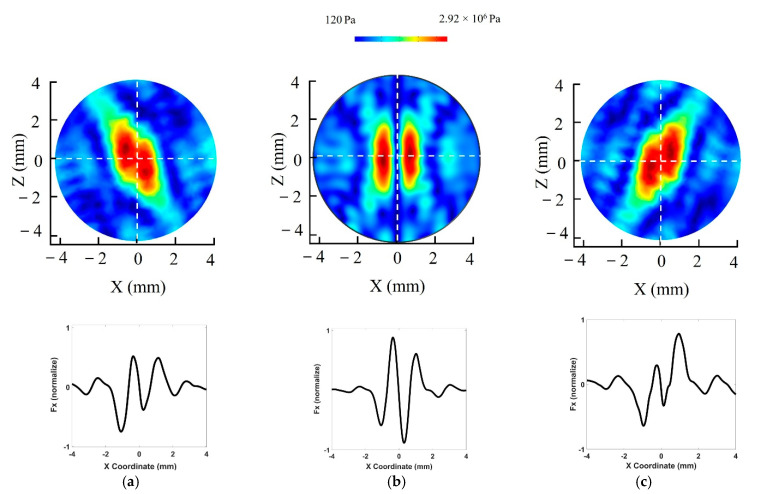
Acoustic pressure field at the trapping point with a 50 Vpp difference between two sides of the tweezer. (**a**) Excitation voltages are 60 and 10 Vpp for the left and right sides, respectively. (**b**) Excitation voltages for both sides of system are 60 Vpp. (**c**) Excitation voltage on the left side is 10 Vpp and that on the right side 60 Vpp.

**Figure 6 pharmaceutics-14-01490-f006:**
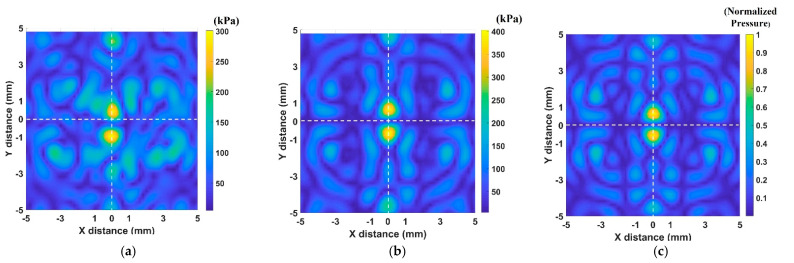
The acoustic pressure field map: (**a**) Scanned acoustic pressure value before calibration; (**b**) Scanned acoustics pressure value after calibration; (**c**) acoustic pressure obtained by simulation.

**Figure 7 pharmaceutics-14-01490-f007:**
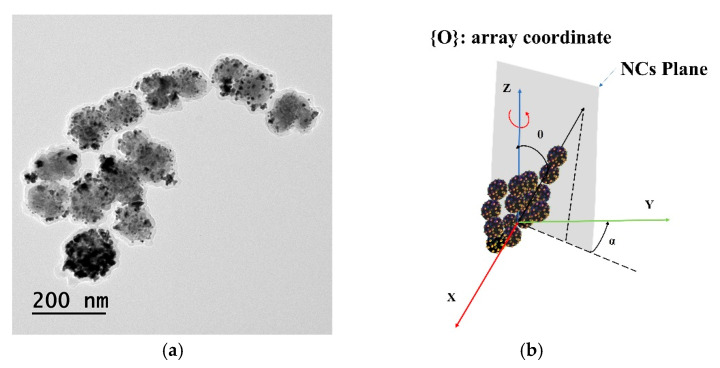
Characteristic of NCs: (**a**) TEM image of NCs; (**b**) The plane of NCs in the {O} coordinate system.

**Figure 8 pharmaceutics-14-01490-f008:**
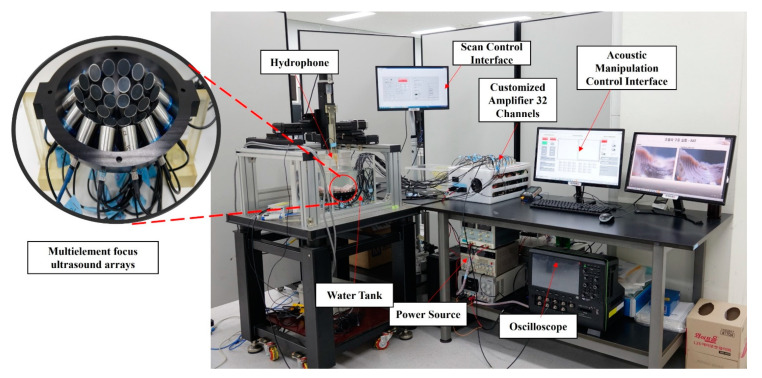
Experimental setup of the acoustic manipulation system for five-DoF manipulation control: List of all components in the AcoMan system (**right**) and fabricated multi-element focus ultrasound arrays (**left**).

**Figure 9 pharmaceutics-14-01490-f009:**
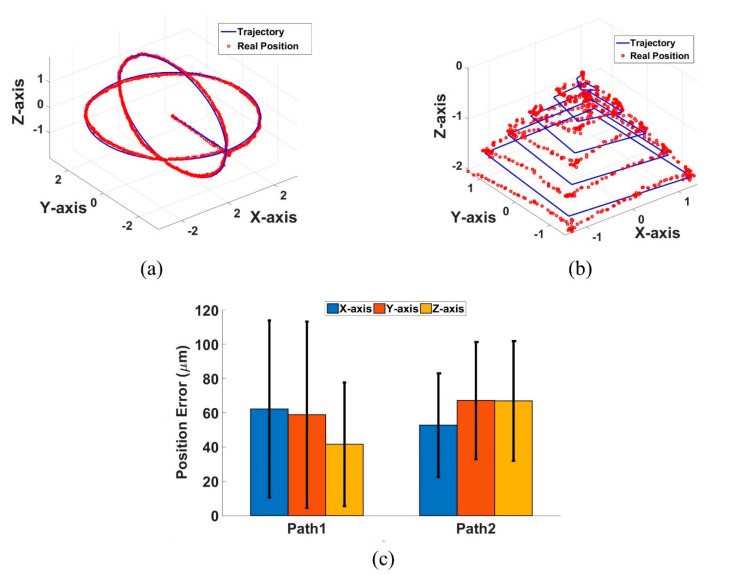
Demonstration of 3D automated close-loop control: (**a**) Trajectory with a globular shape—path 1; (**b**) Trajectory with a pyramid shape—path 2 keeping its orientation constant along the *X*-axis. The way points (red •) are data tracked by the vision system. The green line ((-) is the pre-programmed trajectory for reference; (**c**) Root mean square and standard deviation of position error in the -*X*, -*Y*, -*Z* axis in path 1 and path 2.

**Figure 10 pharmaceutics-14-01490-f010:**
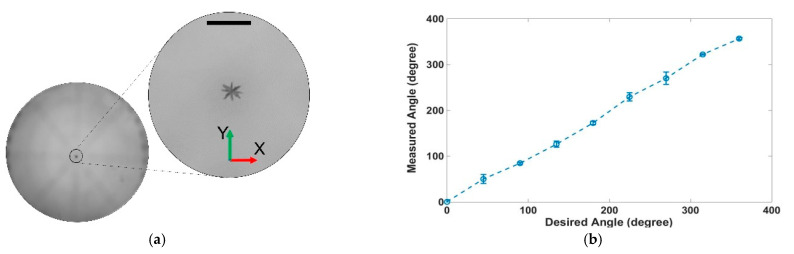
Demonstration of rotation control in the O-XY plane: (**a**) time-lapse image sequences show a nanocarrier cluster in the O-XY plane (viewed from above) rotated at an arbitrary location in space (scale bar is 1 mm); (**b**) difference angle between the measured and desired angles in open-loop control.

**Figure 11 pharmaceutics-14-01490-f011:**
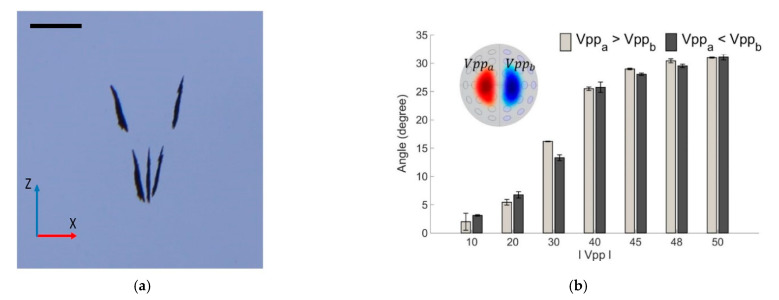
Demonstration of rotation control in the O-XZ plane: (**a**) Time-lapse image sequences show a nanocarrier cluster in the O-XZ plane (viewed from front) rotated along the *X*-axis then keeping the orientation and moving forward (scale bar is 1 mm); (**b**) Counterclockwise and clockwise angles along the *X*-axis for 10 and 50 V between two sides of the tweezers.

**Figure 12 pharmaceutics-14-01490-f012:**
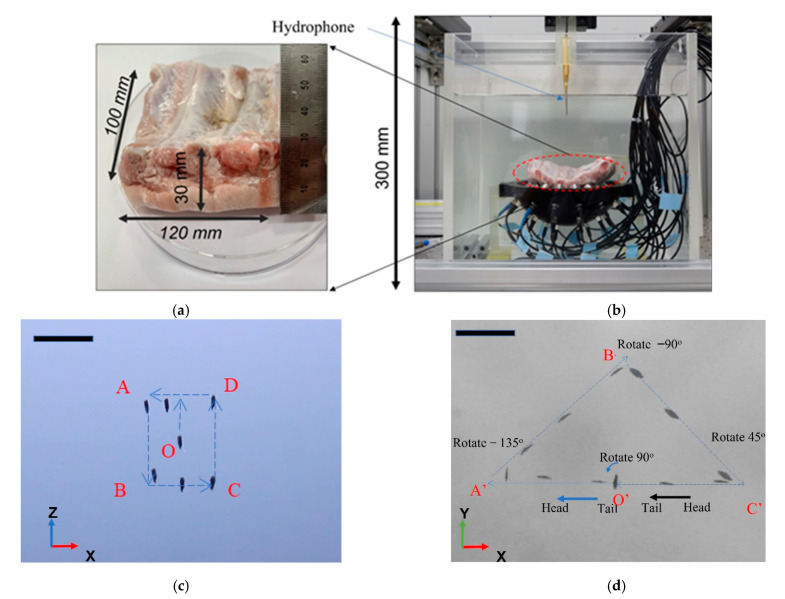
Ex vivo experiments for testing the manipulation and rotation of nanocarrier cluster: (**a**) Porcine rib dimension; (**b**) Experimental setup; (**c**) Time-lapse images of the manipulation in the XOZ planes; (**d**) Time-lapse images of the manipulation and rotation in the XOY plane (scale bar: 2 mm).

**Table 1 pharmaceutics-14-01490-t001:** Comparison between the proposed and recently developed systems.

Acoustic Tweezers Type		Acoustics Actuation System [Reference]	Number of DoF	Objects	Mechanism
Traveling wave tweezers	Active	Proposed system	5 (3 DoF of translation and 2 rotation)	Nanocarrier clusters (600 μm × 300 μm)	Single-side multielement transducers (1 MHz)
		Marzo et al. [4]	4 (3 DoF of translation and 1 rotation)	Polystyrene particles (0.6 to 3.1 mm diameter)	Single-side multielement transducers (40 KHz)
	Passive	Melde et al. [9] Franklin et al. [23]	3 DoF of translation	Droplets (mm scale) Particle (150 μm)	Single transducer with lens (688 KHz)
Standing wave tweezers	Surface acoustic waves	Ding et al. [24]	3 (2 DoF of translation and 1 rotation)	Droplets or cells (Blood cells)	IDT, microfluidics channels
	Bulk acoustic waves	Ghulam et al. [25]	1 Free DoF of rotation	Particle (1–30 μm diameter)	IDT (10, 133 MHz), microfluidics channels
Acoustic-streaming tweezers	Bubble based	Daniel et al. [19] Ali et al. [26]	3 DoF of rotation	Cell, small organisms	PDMS microchannel single transducer (24 KHz)
	Solid structures	Huang et al. [27]	3 DoF of rotation	Polystyrene beads (10 and 0.9 μm)	PDMS microfluidic single transducer (1–100 KHz)

**Table 2 pharmaceutics-14-01490-t002:** Specifications of AcoMan.

Parameter	Value	Unit
Number of elements	30	EA
Resonance frequency	1	MHz
Time delay resolution	25	ns
Controllable Voltage (Vpp)	10 to 200	Vpp
Diameter (D)	75	mm
The distance from array surface to focal point (h)	31	mm

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
