# Peer review of "Holographic Acoustic Tweezers for 5-DoF Manipulation of Nanocarrier Clusters toward Targeted Drug Delivery"

_pharmaceutics, 2022, doi:10.3390/pharmaceutics14071490_

Round 1

Reviewer 1 Report

The authors should re-illustrate the figures

Should add at least 2 master tables to compare the results of the literature

Should add statistical analysis to all of the numerical results

Reviewer 2 Report

The authors have developed an experimental setup that makes it possible to manipulate nano- and micrometer particles in aqueous solutions. In a volume of 5x5x4 mm, particles can be moved to a given point and rotated through a given angle. In the future, this approach may be very promising for practical applications in medicine.

The advantages of the article include the experimental proof of the effectiveness of the approach and the existence of the experimental setup to improve it.

However, the article has a significant number of obvious shortcomings.

There are a large number of formatting errors. Lines 70-71, 147, 231, 255, 285, 292-297, 339, need to be corrected.

The authors do not provide any estimates of the selectivity of the method. Both the text of the article and the title "nanomotor" used for the cluster, give the impression that the size of the manipulated area is larger than a micrometer. A cluster of nanoparticles can be moved as a whole, but can it be broken down into individual particles?

What method was used to determine the spatial position and orientation of the cluster? (Fig. 9, 10, 4a(Line 339), Fig. 12d)

The function amplifier works in the range of 1 Hz to 40 MHz. What is the reason for choosing the operating frequency of 1 MHz?

By how much does the temperature of the medium rises in the focus of the manipulator?

Reviewer 3 Report

This manuscript describes pharmaceutical technology and the methodology of targeted drug delivery of the future. Manuscript is characterized by a high degree of novelty. Acoustic tweezers provide unique capabilities in medical applications, such as contactless manipulation of small objects (e.g., cells, compounds or living things) from nanometer-sized extra-cellular vesicles to centimeter-scale. Additionally, they are capable of being transmitted through the skin to trap and manipulate drug carrier in various media.

Manuscript is very well organized with a lot of schemes that make it easier to keep track of complex topics. The authors tried to explain the complex topic at a level that the average reader of the journal will understand without much effort. The discussion and conclusion are fully supported by experimental results.

Suggestion:

Do the authors have knowledge of how the viscosity of the fluids in which they move (orient) the drug carriers (microrobots) affects the degree of dissipation of acoustic energy used to move the nanoobject. How does the ratio of the total surface area of aggregated nanoobjects and the viscosity of the medium affect the degree of acoustic energy dissipation, whether the temperature of the medium changes during the manipulation of nanoobjects.

Reviewer 4 Report

The authors developed a five degree-of-freedom contactless manipulation acoustic system (AcoMan) using 30 ultrasound phased arrays. Traveling waves are generated to control the position and orientation of a fully untethered nanocarrier cluster (NCs). Phase modulation and switching power supply for each ultrasound transducer are employed to rotate the NCs in the horizontal plane and control the power supply for the rotation in the vertical plane. Both in vitro and ex vivo experiments have been carried out to confirm the performance of this system. This work has great potential in the targeted drug delivery. However, this article in the current version is not in a good shape and quality for publication. Significant revision is required. English also needs significant polish.

Why the rotation is only in O-XY and O-XZ planes, not in O-YZ plane?

In the ex vivo experiment, how to correct the phase for acoustic wave propagation through the heterogeneous media? Especially for the future in vivo application without hydrophone scanning?

In the future in vivo drug delivery, without the two cameras, how to do the control since there will be no feedback images available?

The authors should not emphasize wireless manipulation

Citation errors should be corrected.

Table 1 is not complete, there are many other acoustic tweezers.

Line 20 hard to figure out why UTs can present ultrasound phased arrays

Line 26-27 Ex vivo experiments in porcine ribs?

Line 28 change target drug delivery to targeted drug delivery

Line 33 bad use of made of here

Line 37-38 Robert W. Boyle et al. is not a single person

Line 45 bad expression of micrometer-to-centermeter range

Line 56 clarify parallel multidimensional transducer arrays

Line 57 according to the authors classification, standing wave and acoustic streaming tweezers are different so that what s the meaning of standing waves for acoustic streaming tweezers?

Line 58-59 traveling wave can be produced using a single-element transducer, either focused or unfocused.

Line 81 change OXY plane to O-XY plane, thereafter. Is a 45° step angle resolution too large?

Line 84 amplitude power or power amplitude?

Figure 1. concept of micro/nano robots is wrong here, a passive substance cannot be called robot by the definition. Correct this concept throughout the manuscript.

Line 96 wrong use of few here

Line 101 bad expression of be controlled by controlling

Line 111-112 clarify acceleration control

Line 115 no indentation, thereafter.  P0 is not the power.

Line 116 the meaning of A is not clear

Eq. (2) is wrong

Line 134 w is the angular frequency

Line 137 the use of symbol should be consistent. XYZ has been used to present the Cartesian axes before

Line 144-145 grammar error of with the number of transducers

Figure 2 it shows that the array used in this study is not in hemispherical shape

Table 2 grammar error of the high of focus point

Line 156 grammar error of focus points

Line 157 wrong expression of objects strongest acoustic radiation force

Line 167 very confusing, why the accuracy for current is ±0.1V? thereafter

Eq. (6) is in the wrong position

Line 194-195 awful expression The simulation ... can be simulated

Figure 3 A single focus or a twin in the simulation? Wrong use of manipulation here. Color bar is required for the acoustic pressure field map. Why the acoustic force is negative? Acoustic radiation force at the focus rather than along the array axis is required.

Line 214 This statement is not correct. It should be the number of twin trap signal references rather than the number of transducers.

Line 231-234 wrong figure number, a color bar is required.

Line 237 1 mm needle hydrophone is too large for a good and reliable measurement of the acoustic field since the averaging effect across the sensing element must be considered.

Eq.(8) in the wrong position

Line 255-256 Are they absolute acoustic pressure field or normalized acoustic pressure field?

Line 258-287 They are not Results, but Materials and Methods.

Line 268 It is still unclear how to determine the plane of NCs.

Line 270 set up not setup

Line 291-297 repeated sentences

Line 369 this design is not novel

Line 370 grammar error

Line 371 1 Hz is not ultrasound signal

Round 2

Reviewer 1 Report

Accept

Author Response

Thank you so much for your reviewing and positive decision. 

Reviewer 2 Report

The reviewer's comments have been answered satisfactorily.

Author Response

(The authors gave the same response as above.)

Reviewer 4 Report

English needs professional editing service.

Control the position and orientation of nano-clusters using twin trap should be explained in details. For example, how to optimize the distance between these two foci.

Unnecessary of using two different terms, ultrasound-phased array and ultrasound transducers, to present the same thing.

In the normalization, which value is used as the reference?

Hydrophone is used for the acoustic pressure waveform measurement, not the direct determination of phase.

Presentation format of the plane is not consistent.

There is no explanation of closed-loop control approach.

In the authors response, the question of how to correct the phase aberration through heterogeneous media without the hydrophone measurement in vivo is not answered correctly. Thus, it is not sure the potential of the proposed system for the future clinical application.

Line 22 clarify pointing orientation

Line 25 grammar error of each UTs, thereafter

Line 40 e.g., not e.g..,

Line 43 change in the air to in air

Line 55-56 clarify parallel multidimensional transducer arrays , standing waves can be generated without a reflection layer

Line 58-60 concept in this sentence for travelling wave is not correct

Line 72 in various media? only water is used by the authors

Table 1 microfluidics is not the medium, no in vivo experiment was done using the proposed system

Line 85 clarify the maximum angle of pointing

Line 87 change to In the future

Line 117 specify transducer normal, not all variables in Eq. (1) have been explained here. I found some of them are after Eq. (2). For readers, it is very inconvenient.

Line 124 in the acoustic fields

Line 126-127 hard to understand this sentence, without speed and angle control can control the instability

Line 130-132 concept of this statement is wrong

Line 138-139 Which medium has such properties? Specify acoustic velocity

Line 143-144 penetration depth not depth penetration

Fig. 2 and Table 2 If the array is not in semi-spherical shape, the aperture diameter is not the curvature diameter

Line 160-161 at the focus or at the focal point, thereafter

Line 163 specify flexible amplifier

Line 277-278 awful expression, The designed UT was built with 30 ultrasound transducers.

Fig. 11b no x-axis label

Line 333-334 how to measure the temperature?

Line 351 there is no accepted term of function amplifier

Line 352 1Hz is not high frequency range

Line 382-385 the spatial resolution of sonography is usually 1 mm in the abdominal examination. How can it figure out the nano-clusters for the feedback?

Round 3

Reviewer 4 Report

there is no accepted terminology of "single-side ultrasound phased array"

both UA and UT are used to present the same thing. Although I pointed it out, the authors ignore this error during the revision.

Line 56-58 concept of standing waves is not correct

Line 58-59 wrong expression of "using standing waves for acoustic streaming tweezers"

Line 60-62 I pointed out many times that planar transducer can also generate traveling waves. In addition, traveling-wave tweezers will be affected by the beam,  reflection layer and microfluidic channel.

Line 62-64 no idea why noninvasive therapeutic application rather than tweezers is mentioned here since they are different things.

Table 1. no idea what's the real meaning of "medium" by the authors. particles, cells or small organisms cannot move in the PDMS. Classification is quite confusing. 

117-121, 136-137 change ";" to ","

Line 129-130 explain "without speed and angle control"

Fig. 2 illustration of curvature is not correct.

Line 204 What's the meaning of "focus" here since there are two foci as shown in Fig. 3.

Line 248-249 the focal point is a point, so what is the definition of the center of a point?

Line 262-263 wrong concept of "the normalized absolute acoustic pressure"

Line 297-314 bad writing. Do authors check it before resubmission?

The authors did not discuss how to apply this technology in vivo without camera monitoring for closed-loop control. Therefore, the readers don't know the potential of this work clearly.
